



# Correction of field instabilities in biomolecular solid-state NMR by simultaneous acquisition of a frequency reference

Václav Římal[1], Morgane Callon[1], Alexander A. Malär[1], Riccardo Cadalbert[1], Anahit Torosyan[1], Thomas Wiegand[1], Matthias Ernst[1], Anja Böckmann[2], Beat H. Meier[1]

[1]Physical Chemistry, ETH Zurich, Zurich, CH-8093, Switzerland

[2]Molecular Microbiology and Structural Biochemistry, UMR 5086 CNRS/Université de Lyon, 69367 Lyon, France

*Correspondence*: Beat H. Meier (beme@ethz.ch)

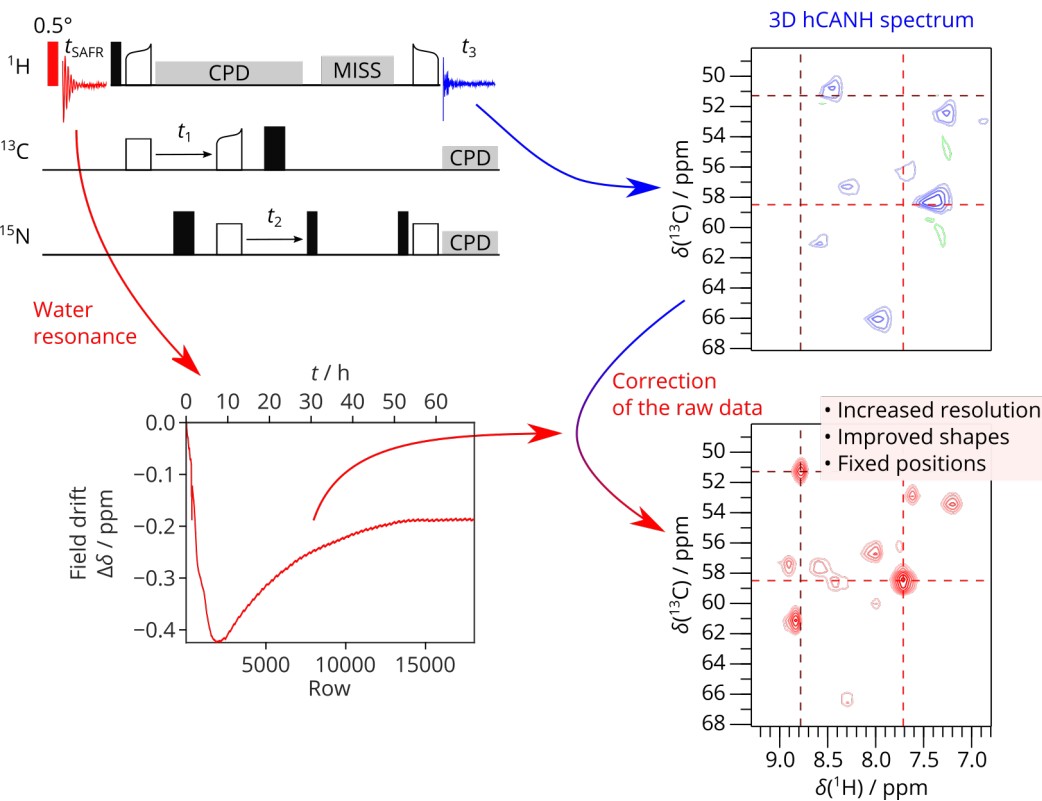





**Abstract.** With the advent of faster magic-angle spinning (MAS) and higher magnetic fields, the resolution of biomolecular solid-state nuclear magnetic resonance (NMR) spectra has been continuously increasing. As a direct consequence, the always narrower spectral lines, especially in proton-detected spectroscopy, are also becoming more sensitive to temporal instabilities of the magnetic field in the sample volume. Field drifts in the order of tenths of ppm occur after probe insertion or tempera-

ture change, during cryogen refill, or are intrinsic to the superconducting high-field magnets, particularly in the months after charging.

As an alternative to a field−frequency lock based on deuterium solvent resonance rarely available for solid-state NMR, we present a strategy to compensate non-linear field drifts using simultaneous acquisition of a frequency reference (SAFR). It is based on the acquisition of an auxiliary 1D spectrum in each scan of the experiment. Typically, a small-flip-angle pulse is

added at the beginning of the pulse sequence. Based on the frequency of the maximum of the solvent signal, the field evolution in time is reconstructed and used to correct the raw data after acquisition, thereby acting in its principle as a digital lock system. The general applicability of our approach is demonstrated on 2D and 3D protein spectra during various situations with a non-linear field drift. SAFR with small-flip-angle pulses causes no significant loss in sensitivity or increase in experimental time in protein spectroscopy. The correction leads to the possibility of recording high-quality spectra in a typical

biomolecular experiment even during non-linear field changes in the order of 0.1 ppm h$^{-1}$ without the need for hardware solutions, such as stabilizing the temperature of the magnet bore. The improvement of linewidths and peak shapes turns out to be especially important for $^{1}$H-detected spectra under fast MAS, but the method is suitable for the detection of carbon or other nuclei as well.

## 1        Introduction

Solid-state nuclear magnetic resonance (NMR) witnesses an ongoing increase in spectral resolution, in particular in biomolecular applications, and proton linewidths in the order of 10 Hz (or 0.01 ppm at 1200 MHz) are possible in $^{1}$H-detected spectra under fast magic-angle spinning (MAS) for perdeuterated and fully back-exchanged proteins (Penzel et al., 2019). To fully exploit this high resolution, crucial to extract as much information as possible, the static magnetic field $B_0$ must be stable within 1 ppb (part per billion) during the duration of the entire experiment (minutes to days). Otherwise, ade-

quate corrections must be applied. The stability is particularly critical at a high magnetic field (Callon et al., 2021; Nimerovsky et al., 2021). Alternatively to correcting the field, the spectrometer frequency could be adjusted in real time. While this is conceivable with modern hardware, we are not aware of such a solution in biomolecular NMR and significant protocol changes would be necessary.

In solution-state NMR, the $B_0$ instabilities are routinely compensated for by a deuterium field−frequency lock in real time. In

solid-state NMR spectroscopy, the implementation of a lock proved difficult in practice and commercial probeheads are rarely equipped with a lock channel. Modern actively shielded superconducting magnets are reasonably stable in a magneti-



cally tranquil environment and at a constant laboratory temperature with the highest change often given by a linear drift of the field as a function of time. Spectrometers typically provide a hardware linear drift correction that compensates for this field drift, which can be adjusted every few weeks or months by a calibration experiment. More importantly, insertion and
removal of the NMR probe from the magnet bore and any changes in the temperature of the sample or the shim cylinder create a relatively strong non-linear drift that lasts, for high-field magnets, for hours (Malär et al., 2021), delaying the start of an experiment.

Alternatives to the field−frequency lock approaches, that are applicable to solid-state NMR, too, have been used for reducing the field drift. Malär et al. (2021) describe a spectrometer equipped with a magnet-bore heater system that prevents major
long-term field drifts due to temperature changes inside the magnet bore, which greatly increases stability and shortens time constants to reach a stable field after a disturbance. External locks can monitor the $^2$D or $^7$Li resonance frequency of an auxiliary sample located in the proximity of the main sample, which is then utilized for the lock (Paulson and Zilm, 2005; Takahashi et al., 2012). On benchtop low-field NMR spectrometers, a "SoftLock" system monitoring the non-deuterated solvent signal is available (Minkler et al., 2020). In magnetic resonance imaging, the field−frequency lock (Henry et al., 1999) has
advanced to feedback control based on field mapping (Vionnet et al., 2021).

Another solution that eliminates the effect of the field drift in the spectra is data correction after the acquisition. Referencing to an internal $^{13}$C resonance was documented (Kupče and Freeman, 2010). Interleaved navigator scans (Thiel et al., 2002) or spectral registration (Near et al., 2015) can be used in medical magnetic resonance spectroscopy in vivo. More recently, a posteriori linear drift correction was implemented for solid-state NMR spectroscopy (Najbauer and Andreas, 2019). The lat-
ter method profits from a fast and easy application without any requirement of extra hardware, because one only needs to measure a reference spectrum before and after a multidimensional experiment. However, the application is restricted to linear drifts during the entire experiment.

Our work presented here extends the linear drift compensation (Najbauer and Andreas, 2019) for a general non-linear case. This is necessary especially for experiments started several hours after sample insertion or after or during refill of liquid he-
lium or nitrogen to the magnet, when long-lasting field perturbations emerge. We propose to monitor the magnetic field for each trace of a multidimensional experiment. On current spectrometers, the field monitoring can proceed as simultaneous acquisition of a frequency reference (SAFR): a reference spectrum with a small-flip-angle pulse is recorded before each scan of the experiment. For high-field magnets, the flip angle can be chosen short enough (we used 1° or less) not to cause any significant impact on the rest of the experiment. Simultaneous acquisition of several free-induction decays (FIDs) in one scan,
as in SAFR, is becoming increasingly popular (Gallo et al., 2019; Gopinath and Veglia, 2020; Stanek et al., 2020; Kupče and Claridge, 2017; Sharma et al., 2016). We present various examples of $^1$H- and $^{13}$C-detected 2D and 3D MAS spectra that benefit from SAFR and the subsequent correction of the multidimensional FID. The pulse sequences and the correction program are published along with this article (Římal, 2021).



## 2 Theory

### 2.1 Phase differences during unstable magnetic field

All our treatment deals with phase-sensitive data, both in direct and indirect dimensions of a multidimensional experiment. In a magnetic field with a time-independent magnitude $B_0$, the NMR signal in the time domain (defined by the variable $t_{acq}$) is expressed as

$$S(t_{acq}) = S_x(t_{acq}) + i S_y(t_{acq}) ,$$ (1)

where we identify $x$ and $y$ with the real and imaginary axes, respectively. Besides the direct acquisition of an FID of an isotope with a gyromagnetic ratio γ ($t_{acq} = t_{dir}$, in a 2D typically $t_2$), Eq. (1) applies to any indirect evolution block of an $n$-dimensional experiment ($t_{acq} = t_{indir}$, in a 2D typically $t_1$).

If the magnetic field changes in time, the detected coherence acquires an additional phase $\Delta\varphi$ relative to the evolution in the constant field $B_0$ and the detected values are then $S_x^*(t_{acq})$ and $S_y^*(t_{acq})$. In the following, we describe how the ideal complex $S(t_{acq})$ from Eq. (1) can be reconstructed from $S_x^*(t_{acq})$ and $S_y^*(t_{acq})$ given that the time dependence of the magnetic field is known – either measured by SAFR as proposed in this work or obtained from another source.

### 2.2 Correction of the direct dimension

We assume that the field change is slow compared to the direct acquisition period, hence the $k$-th FID of a multidimensional experiment is recorded under a constant field with the magnetic induction $B_0 + \Delta B_k$. The phase difference of the time-domain signal depends linearly on time $t_{acq} = t_{dir}$:

$$\Delta\varphi_k(t_{dir}) = \gamma \Delta B_k t_{dir} .$$ (2)

Therefore, the complex experimental data can be corrected using rotations by $-\Delta\varphi_k(t_{dir})$ for each time point in $t_{dir}$:

$$S(t_{dir}) = S^*(t_{dir}) e^{-i\Delta\varphi_k(t_{dir})} = S^*(t_{dir}) e^{-i\gamma\Delta B_k t_{dir}} ,$$ (3)

where $S_x^*(t_{dir})$ and $S_y^*(t_{dir})$ have been combined into a complex FID:

$$S^*(t_{dir}) = S_x^*(t_{dir}) + i S_y^*(t_{dir}) .$$ (4)

### 2.3 Correction of the indirect dimensions

Even after the corrections are done in the direct dimension, one still needs to compensate for the effect of the field change $\Delta B_k$ on the spin evolution during all the indirect time periods present in the pulse sequence. For one indirect dimension, there is a particular value of the evolution delay $t_{indir} = q\Delta t_{indir}$, which is incremented stepwise within the experiment, connected



with the FID $k$ describing the increment $q$ in the indirect dimension. In general, every FID $k$ has a different value of $\Delta B_k$. Because of that, the phase difference of every acquired data point must be treated with its own $\Delta B_k$, such that

$$\Delta\varphi_k' = \gamma_{\text{indir}}\Delta B_k t_{\text{indir}} = \gamma_{\text{indir}}\Delta B_k q\Delta t_{\text{indir}} \ , \tag{5}$$

where $\gamma_{\text{indir}}$ is the gyromagnetic ratio of the isotope evolving during the indirect evolution period.

In order to be able to rotate the acquired data back to the corrected points $S(t_{\text{indir}})$ accurately, it is important to realize that different fields will be present during the acquisition of the real and imaginary parts of the data that form a hypercomplex point. This can be particularly important for adjacent planes in 3D experiments and yet more for hyperplanes in higher dimensions, where significant wall-clock time has elapsed between the respective acquisitions. According to Eq. (5), a pair of the detected values $S_x^*(t_{\text{indir}})$ and $S_y^*(t_{\text{indir}})$ thus carries unequal phase differences $\Delta\varphi_k'$ and $\Delta\varphi_l'$, respectively, because they are recorded during FIDs $k$ and $l$. Therefore, $S_x^*(t_{\text{indir}})$ and $S_y^*(t_{\text{indir}})$ no longer belong to a single complex number as for the direct dimension in Eq. (4), but are $x$ and $y$ coordinates of two points $S_k^*$ and $S_l^*$, respectively, which are images of $S$ rotated by angles $\Delta\varphi_k'$ and $\Delta\varphi_l'$ (Fig. 1). For States and States−TPPI (time-proportional phase incrementation) modes (States et al., 1982; Bodenhausen et al., 1984) with $t_{\text{indir}}$ (and $q$ as well) being the same for $k$ and $l$, however, their amplitude is maintained (i.e., $|S_k^*|=|S_l^*|$). $S_x^*$ and $S_y^*$ can be viewed as the coordinates of the point $S$ in a frame with $x$-axis rotated by $-\Delta\varphi_k'$ and $y$-axis by $-\Delta\varphi_l'$, which is obtained after a transformation by a generally non-orthogonal matrix

$$\mathbf{R} = \begin{pmatrix} \cos\Delta\varphi_k' & -\sin\Delta\varphi_k' \\ \sin\Delta\varphi_l' & \cos\Delta\varphi_l' \end{pmatrix} . \tag{6}$$

The desired coordinates of the point $S$ in the original frame, i. e., the values corrected for the field drift, are then obtained by performing a back-transformation:

$$\begin{pmatrix} S_x \\ S_y \end{pmatrix} = \mathbf{R}^{-1}\begin{pmatrix} S_x^* \\ S_y^* \end{pmatrix} , \tag{7}$$

where

$$\mathbf{R}^{-1} = \frac{1}{\cos\Delta\varphi_k'\cos\Delta\varphi_l' + \sin\Delta\varphi_k'\sin\Delta\varphi_l'}\begin{pmatrix} \cos\Delta\varphi_l' & \sin\Delta\varphi_k' \\ -\sin\Delta\varphi_l' & \cos\Delta\varphi_k' \end{pmatrix} . \tag{8}$$

In the special case of $\Delta\varphi_k'=\Delta\varphi_l'$, Eq. (8) reduces to a single orthogonal rotation by $-\Delta\varphi_k'$, consistently with Eq. (3) valid for the direct dimension.

For TPPI, $S_k^*$ and $S_l^*$ are acquired with different values of $t_{\text{indir}}$, which renders the above approach invalid. A workaround for TPPI is to convert the data to States−TPPI first, which is possible by forward and backward Fourier transforms with appropriate modes. Without this, mirrored signals appear in the corrected spectra.



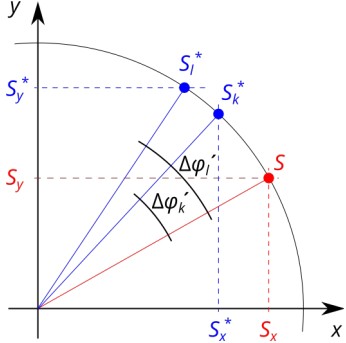

**Figure 1: Without the magnetic-field drift, the values $S_x$ and $S_y$, which form the Cartesian coordinates of the point $S$, would be measured as the phase-sensitive signal in the indirect dimension under States or States–TPPI mode (red). During an experiment** 130 **with a time-dependent magnetic field, the $x$ and $y$ coordinates of the points $S_k^*$ and $S_l^*$ are acquired, respectively (blue). $S_k^*$ and $S_l^*$ are images of the point $S$ after rotations by $\Delta\varphi_k$' and $\Delta\varphi_l$', respectively.**

## 3    Materials and methods

### 3.1    Samples

Uniformly $^{13}C-^{15}N$ labelled PYRIN-domain filaments of mouse apoptosis-associated speck-like protein containing a cas-
135 pase-recruitment domain (ASC) were expressed in *E. coli* as described previously (Ravotti et al., 2016). A deuterated and
100 % back-exchanged ASC sample (dASC) was prepared in the same way except that deuterated water, glucose, and am-
monium chloride were used. Exchangeable sites were then re-protonated by solvent exchange with $H_2O$. Published protocols
for preparation of uniformly $^{13}C-^{15}N$ labelled samples were followed for Rpo4/7*, a complex of two subunits of RNA poly-
merase II Rpo4C36S/Rpo7K123C (Torosyan et al., 2019; Werner and Grohmann, 2011), and for HET-s(218–289) amyloid
fibrils from the filamentous fungus *Podospora anserina* (Van Melckebeke et al., 2010).

### 3.2    Solid-state NMR spectroscopy

The NMR experiments were performed using Bruker triple-resonance MAS probeheads on two wide-bore 850 MHz Bruker
Avance III HD (running Topspin 3 software) and a standard-bore 1200 MHz Bruker Avance NEO (Topspin 4) spectrometers.
$^1H$-detected spectra were acquired using 0.7 mm rotors at a 100 kHz MAS rate. $^{13}C$-detected spectra were acquired using 3.2
145 mm rotors in E-free probes (Gor'kov et al., 2007). The MAS rates for HET-s(218–289) were 17 kHz and 20 kHz for 850
MHz and 1200 MHz, respectively; the MAS rate for experiments on $^{13}C$-adamantane was 11 kHz. All spectra were processed
in Topspin 4 and presented using CcpNmr Analysis 2 (Stevens et al., 2011, Vranken et al., 2005). Basic acquisition and pro-
cessing parameters are shown in Tables S2 and S3 in the Supplement.



### 3.3 Pulse programs with SAFR

The SAFR block usually consists of a small-flip-angle pulse on protons and the acquisition of an FID preceding the main experiment (Fig. 2). It is repeated every scan and summed up and stored to disk at the same time as the data of the main experiment, i. e., before the acquisition of the next row of a multidimensional experiment is started. No differences in total experimental time and longitudinal relaxation to equilibrium are introduced since the entire SAFR block is placed during the relaxation delay and the small-flip-angle pulse does not disturb the equilibrium polarization noticeably. There are options in the

pulse programs, which switch the SAFR block off (keeping the overall timing unchanged), include decoupling on an additional channel, or acquire a one-dimensional spectrum by the same pulse program. We distinguish homonuclear and heteronuclear SAFR with respect to the nucleus detected in the main experiment. Homonuclear variants work with one receiver and use separated memory buffers (tested on Bruker Avance III HD and NEO, but should be possible on most currently used spectrometers). In contrast, heteronuclear cases need hardware that allows acquisition on multiple receivers, which is possi-

ble with modern commercial spectrometers (default on Bruker Avance NEO).

Representative examples of homonuclear, $^{1}$H-detected pulse sequences are 2D SAFR−hNH and SAFR−hCH correlations with cross-polarization (CP) transfers and MISSISSIPPI water suppression (Zhou and Rienstra, 2008) shown in Fig. 2 (a). A small-flip-angle pulse (we used 0.5° or 1.0°) in the SAFR block, which minimizes signal losses in the main part (Fig. S1 in the Supplement), is sufficient for detecting the water signal in a protein sample. The equivalence of 2D experiments with and

165 without SAFR has been tested (Fig. S2 in the Supplement). Schemes of 2D SAFR−hNH and SAFR−hCH with INEPT transfers and 3D SAFR−hCNH pulse programs can be found in the Supplement (Fig. S3 and Fig. S4, respectively). Even a combination of SAFR with multiple acquisition periods in the main experiment, based on dual-acquisition MAS (DUMAS) hCH−hNH after excitation by simultaneous CP (Gopinath and Veglia, 2020), has been prepared and tested (Fig. S5 in the Supplement). Despite its limited applicability, we also implemented an X-detected SAFR for X-detected experiments, which

can be run when multiple receivers are not available (Fig. S6 in the Supplement).



(a)

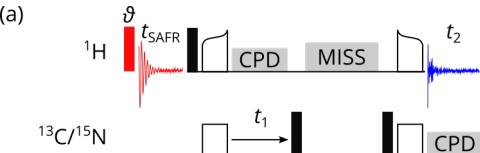

(b)

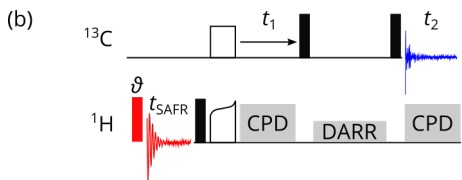

**Figure 2: Pulse programs for ¹H-detected SAFR (red, flip angle $\vartheta$) before a 2D correlation experiment. (a) CP-based ¹H-detected 2D hCH or hNH. (b) 2D ¹³C−¹³C DARR. Filled black rectangles represent 90-degree pulses, empty rectangles and curved shapes are CP transfers, and grey blocks indicate composite-pulse decoupling (CPD), MISSISSIPPI water suppression (MISS), and the DARR. Optionally, decoupling of the third nucleus can be turned on during $t_1$ and $t_2$ in (a).**

When multiple receivers allow detection on two channels in one pulse sequence, X-detected experiments can be accompanied with SAFR on ¹H. Such a heteronuclear case is demonstrated on a ¹³C-detected 2D SAFR−DARR, dipolar-assisted rotational resonance (Takegoshi et al., 2001; Takegoshi et al., 2003), in Fig. 2 (b). The same principles as for homonuclear SAFR described above apply. A simple ¹³C-detected 2D ¹H−¹³C correlation and a ¹³C-detected 3D hNCC involving a ¹³C DREAM, dipolar recoupling enhanced by amplitude modulation (Verel et al., 2001), are shown in Figures S7 and S8 in the Supplement, respectively, both with ¹H SAFR. ¹H-detected 3D SAFR−hCAco[C, NH] experiment with ¹H SAFR including an additional ¹³C acquisition yielding a 2D CACO correlation (Gallo et al., 2019) was also tested (Fig. S9 in the Supplement).

### 3.4 Data correction

For the calculations described in Sect. 2, we have written an AU program "safrcorr". The code builds upon the grounds laid for the linear drift compensation by Najbauer and Andreas (2019) with permission from the authors. It can be run directly in Topspin (Bruker), but its core is in C++, allowing modifications to other data formats. The safrcorr program reads the Fourier-transformed reference spectra and finds the global maximum of each. The peak position is refined by a parabolic interpolation through three adjacent data points (Press et al., 2016). In this way, the information about the field drift $\Delta B_k$ is obtained for every FID of the main experiment. The time-domain data of the main experiment are then corrected in all dimensions according to Eqs. (3) and (7) and saved to disk under a new experiment number. The chemical shifts of the spectral maxima and their frequency differences relative to the first FID are also displayed in a window and stored in a plain-text file. The time needed for the correction is only a few seconds even for spectra with more than 10000 FIDs. The typical disk space occupied by all data after the correction reaches five times the size of the same experiment without SAFR (the main experi-





ment and the SAFR data acquired, their copies after they are split into two individual datasets, and the corrected data), but most of the processed spectra and also the raw data after the split can be safely deleted to free some storage space. Further description of the program and its use in connection with the SAFR pulse programs can be found in the Supplement.

## 4 Results

We present here a collection of $^1$H- and $^{13}$C-detected protein 2D and 3D spectra with SAFR and show the effect of the correction of the field drift. We focus on the non-linear time dependence of the magnetic field. Although strong non-linearities were encountered in the natural drift of the 1.2 GHz magnet several months after its installation (Fig. S10 in the Supplement), the drift at later stages when SAFR was developed showed mostly linear trends during reasonably long experiments. Nevertheless, SAFR is valuable in connection with additional perturbations usually arising, e. g. sample change in probe-

heads that need to be removed, probehead change, sample-temperature changes, the refill of cryogenic liquids for the magnet, or environmental magnetic field changes (possibly coming from other devices in the proximity of the magnet). Even though parts of these field changes can be reduced by other means, such as a bore-temperature control system, it would require dedicated hardware that can be expensive or unavailable and imperfect; SAFR offers a software solution that is independent of the source and time-course of the perturbations because it uses the information obtained directly from the sample

space. In addition, we demonstrate the performance of SAFR during intentionally introduced strong field changes by manipulations with the $Z^0$ shim current, which serve as a proof of concept of the new method and its general applicability.

### 4.1 Compensation of thermal effects

Sample cooling is needed to compensate for the temperature rises by friction during fast-MAS experiments and a temperature of the input gas as low as 240 K is needed. Even after reaching an equilibrium within the sample, which can be moni-

tored by the $^1$H chemical shifts of $H_2O$ in biological samples (Böckmann et al., 2009; Gottlieb et al., 1997), it can take hours until the body of the probehead and the shim cylinder, as well as the magnet bore, reach their thermal equilibrium temperatures without a system that maintains the temperature of these parts (Malär et al., 2021). A similar effect occurs when a change in the sample temperature is required. Due to the temperature dependence of the magnetic susceptibility and the thermal expansion of the materials used, these cooling or heating processes are inevitably connected with a drift of the magnetic

field. Again, the equilibration can take hours without a magnet-bore heater system (Malär et al., 2021).

As an example, we measured a 2D SAFR−hNH spectrum of the per-deuterated PYRIN-domain of apoptosis-associated speck-like protein (Sborgi et al., 2015; Ravotti et al., 2016), dASC, 91 residues, shortly after cooling of the incoming nitrogen gas from 280 K to 240 K had started. The bore-temperature control system, present and functional on the setup used, was disabled during this procedure. SAFR with a 0.5° flip angle was used to determine the resonance frequency of water in



the sample. The signal of SAFR is strong enough after a 0.5° pulse and shows that only a negligible part of the protein amide magnetization is excited (see Fig. 3 for a comparison of SAFR and spin-echo 1D $^1$H spectra). The correction of the field drift of more than 0.1 ppm clearly improves spectral resolution in both dimensions (Fig. 4). Triangular line shapes that are present in the uncorrected spectrum acquired during the unstable conditions are eliminated and the spectral resolution is enhanced. After the correction, the differences to control spectra recorded under constant field (7 and 20 hours later, when the hardware

components have reached their thermal equilibrium) are insignificant and in the same range as between each of the two control experiments performed (Fig. S11 in the Supplement). In this way, employing SAFR to compensate the temperature instabilities allows shortening the delay needed, after setup of the spectrometer or a sample change, before a high-quality spectrum can be acquired, without the requirement of additional hardware equipment such as a magnet-bore temperature control system. Compared to the linear drift correction, no assumptions on the linearity of the time-dependence of the magnetic field

are necessary.



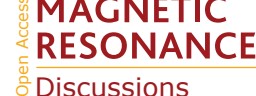

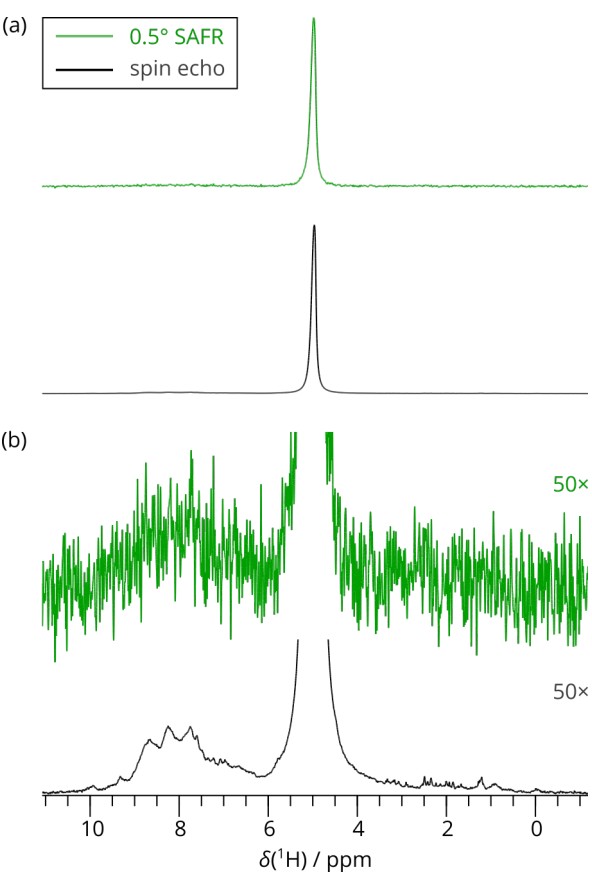

**Figure 3: 1D $^1$H spectra of dASC acquired by SAFR after a 0.5° flip angle during hNH (green) and by a spin-echo pulse sequence (black). 256 scans were recorded in both cases (850 MHz). (a) Full spectra scaled to match their maximal intensities. (b) The spectra in (a) multiplied 50-fold in intensities, showing negligible excitation of the amide region in the 0.5° SAFR spectrum.**

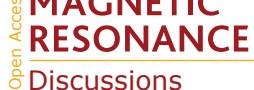




**Figure 4: 2D SAFR−hNH of dASC 36 min after the start of cooling down (850 MHz). (a) The spectrum without (blue) and with (red) the field-drift correction. 1D traces along the horizontal dashed lines are shown at the top. (b), (c) Two selected details of the spectral regions indicated by the dashed rectangles in (a). (d) The evolution of the proton frequency correction.**





### 4.2      2D acquisition during helium fill at 850 MHz

2D hNH CP experiments were acquired on dASC during the helium filling on an 850 MHz magnet. SAFR using the water resonance of the sample was used to monitor the field drift. The 2D experiment was recorded in four identical blocks of eight scans and the 2D FIDs were summed up before Fourier transform. We demonstrate that even when the individual experimental times were set relatively short (1 h 23 min), non-linearities in the field evolution occur (Fig. 5). The drift distorts the line shapes as well as the positions of the resonances, which are restored after the correction (Fig. 5, panel b; comparison to a

control experiment is shown in Fig. S12 in the Supplement).

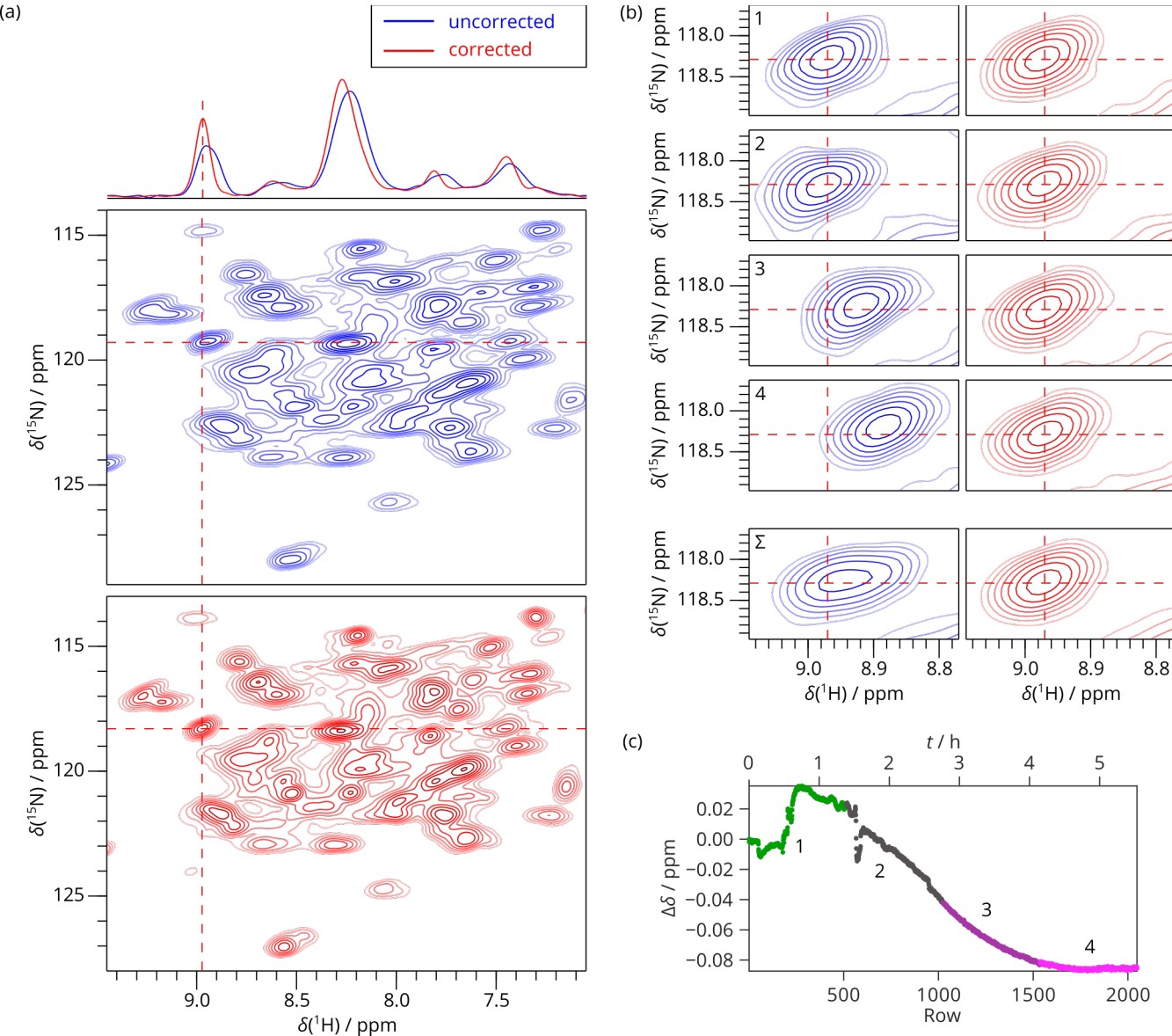

**Figure 5: 2D SAFR−hNH of dASC during helium fill (850 MHz). (a) The spectrum obtained from summing raw data (before Fourier transform) of four experiments together before (blue) and after drift correction (red). 1D traces along the horizontal**
**dashed lines are shown at the top. (b) A selected peak, indicated by the dashed lines in (a), shown separately for the 2D spectra numbered 1−4 and their sum (Σ) before (blue) and after the drift correction (red). Contours in all the panels are plotted at the same intensities per scan. (c) The frequency evolution in terms of $H_2O$ chemical-shift difference measured by SAFR. Different colours correspond to the separate 2D experiments 1–4 (each taking 1 h 23 min with 512 rows, 8 scans per row). The magnet was depressurized around row 50, filling started after row 220, and ended before row 600.**





### 4.3    3D acquisition during helium fill on a 1200 MHz magnet

The performance of SAFR was further exploited during a 3D experiment. Figure 6 shows a CP-based 3D SAFR−hCANH spectrum of dASC (pulse program in Fig. S4 in the Supplement) in a 1200 MHz magnet during helium fill. The improvement after the drift correction is obvious. The distortions in the spectrum before the correction would make it useless which is explained by the strong field drift observed in Fig. 6 (b). Note that the field drift stayed significant for two days; without SAFR, this instrument time couldn't have been used for a reliable 3D. The strongest corrections are along the $^{13}$C axis, which corresponds to the evolution time that is incremented stepwise after a particular NH plane is acquired (the outermost loop in the pulse program). The periodic oscillations seen in the inset of Fig. 6 (b) are not caused by the actual experiment, but were inherent to the magnet control system at that time and were later improved by the manufacturer.

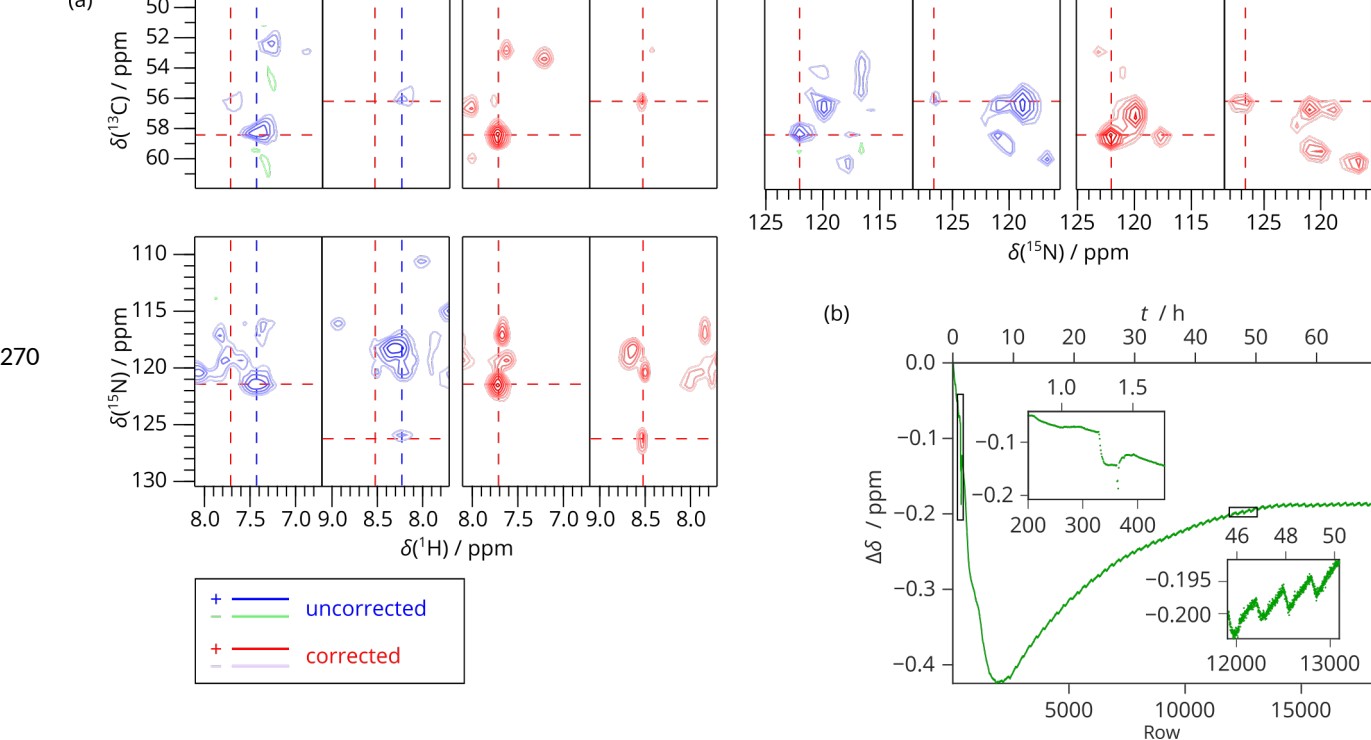

**Figure 6: 3D SAFR−hCANH of dASC during and after helium fill (1200 MHz). (a) Two selected peaks and their surroundings viewed in three possible plane orientations before (blue) and after the drift correction (red). The perpendicular cross-sections are taken at the positions of the peaks after the correction, indicated by the dashed red lines. Differently to this, the $^{15}$N−$^{13}$C plane of the uncorrected spectrum is taken along the dashed blue lines. (b) The frequency evolution in terms of H$_2$O chemical shift difference measured by SAFR. Insets show details of the regions marked by rectangles (different aspect ratios). The helium filling started before the acquisition began.**





### 4.4 2D acquisition during nitrogen fill at 1200 MHz

The 2D SAFR−hCH of a complex of two subunits of archaeal RNA polymerase II (Torosyan et al., 2019), Rpo4/7*, where only Rpo7 (187 residues) was $^{13}$C and $^{15}$N labelled, in Fig. 7 shows an application where no strong field drift was anticipated

but turned out to be present during the experiment. Generally, it is considered safe to measure solid-state spectra during the refill of liquid nitrogen, but a field change of around 0.1 ppm was observed. Because it happened mostly in the second half of the acquisition time, its overall influence on the spectrum is not that strong. Nevertheless, the narrowest peaks are affected, as amplified by the strongly resolution-enhanced processing in Fig. 7: errors in resonance positions in multiples of 0.01 ppm and even false peak doubling (panel b of Fig. 7) can arise when the drift is not compensated for.


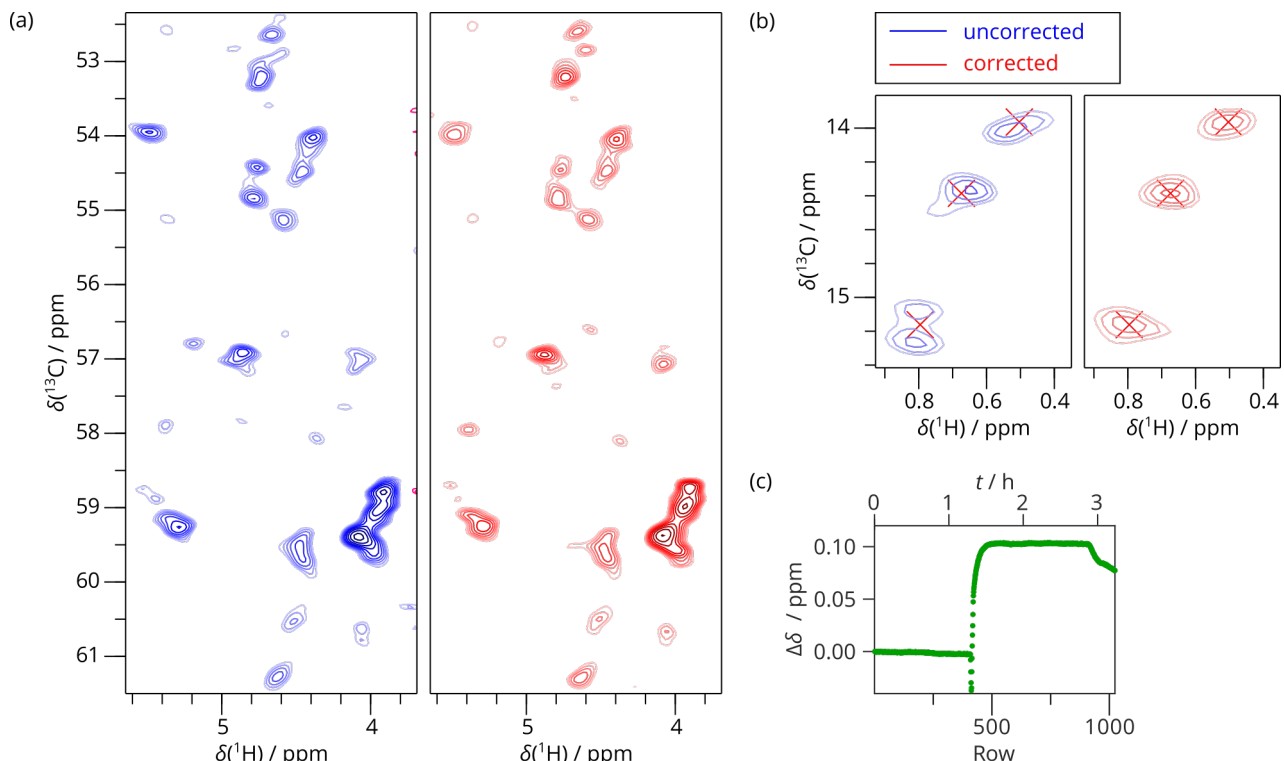

**Figure 7: 2D SAFR−hCH of Rpo4/7* during nitrogen fill (1200 MHz) with processing-enhanced resolution (shifted squared sine bell with parameter SSB = 5 in both dimensions). (a) $C_\alpha$−$H_\alpha$ region of 2D hCH spectrum before (blue) and after the drift correction (red). (b) A detail of the 2D hCH spectrum outside of the region shown in (a). Peak maxima of the corrected spectrum are**
**marked by red crosses. (c) The frequency evolution in terms of $H_2O$ chemical-shift difference measured by SAFR. The nitrogen filling started around row 400.**



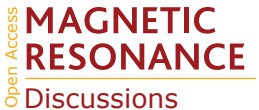

### 4.5 Limitations

The correction scheme assumes that the solvent (water) chemical shift is constant. It is however known that it is temperature-dependent and changes by 0.0111 ppm/°C (Gottlieb et al., 1997), or 13.3 Hz/°C on a 1200 MHz system. The temperature of

the sample must therefore be kept constant within a fraction of a degree (or several degrees, depending on the spectral linewidths) to avoid broadening effects or $t_1$ noise. Of course, similar requirements apply to the conventional field–frequency locks. We have checked that the application of $^1$H-SAFR and subsequent drift correction under ordinary conditions do not introduce extra broadening or noise (Fig. S13 in the Supplement) and that a purely linear field drift is compensated for without any differences to the linear drift correction previously published by Najbauer and Andreas (2019), as shown in the

Supplement (Fig. S14).

Although we expect that the drift correction is mostly needed for $^1$H-detected spectroscopy, there might be relatively rare cases of its usage accompanying X-nucleus detection. $^1$H-SAFR can also be used, but only on spectrometers capable of detection on multiple receivers during one experiment. For single-receiver equipment, $^{13}$C-SAFR is in principle possible for strong and non-linear field changes but can lead to $t_1$ noise after a correction relying on a broad reference resonance (Figures

S15, S16, and S17 in the Supplement).

Furthermore, we assume that the perturbations to be corrected are slow and no changes occur within the duration of a single FID. In addition, we corrected each FID of an experiment (summed up after several scans needed for phase cycling and improvement of the signal-to-noise ratio) as a whole entity. At the cost of higher storage demand and additional pre-processing steps, each scan can be corrected individually in principle, but this did not turn out to be necessary (Figures S18 and S19 in

the Supplement show an extreme case with an artificially created jump in the field value).

### 5 Discussion and conclusions

We have presented a new strategy to eliminate line-broadening and spectral artefacts arising from the instabilities of the magnetic field in high-resolution solid-state biomolecular spectroscopy. By adding a small-flip-angle experiment at the beginning of the pulse sequence, we obtain a frequency reference (SAFR) which can be used to correct digitally the FIDs. An

automated procedure facilitating these steps as well as some example pulse programs commonly employed in biological solid-state NMR are developed and made available. In its principle and outcome, SAFR is equivalent to a lock system: it detects the position of the solvent (or other intense and preferably narrow) peak and corrects the acquired data for the corresponding frequency shift. The main difference is that the correction using SAFR is performed after, not during the acquisition. Our work has demonstrated the importance of a lock in various 2D and 3D correlation experiments under different con-

ditions: an often encountered challenge is the insertion of a sample to the probehead and its temperature change, which can



**MAGNETIC RESONANCE**
Open Access Discussions

delay the start of a conventional acquisition by several hours. SAFR overcomes this problem without the need for a hardware solution, such as a bore-temperature control system. Next, the filling of the cryogenic liquids into the magnet induces non-linear transient field disturbances, which last for hours or even days. Again, their effects on the NMR spectra can be corrected by SAFR. Finally, the field strength is affected by external fields or temperature changes in the room that are unpre-
dictable, but SAFR will be helpful in these cases as well.

SAFR requires a well-resolved and intense resonance line in the spectrum. The [1]H resonance of solvent water present in hydrated microcrystalline or sedimented protein samples typically serves this purpose (Böckmann et al., 2009; Lacabanne et al., 2019). Alternatively, any strong and sufficiently narrow peak can be used as the reference. As the water resonance is temperature dependent, the sample temperature needs to be stable during the experiments within about 1 °C. Normally, one FID
(one row of a multidimensional experiment with its number of scans) is corrected with a single frequency value. If a significant field fluctuation appears within this time, one could store the SAFR spectra for every scan (or a chosen number of scans) and apply SAFR to this entity. We have not implemented this as it seems to be of marginal interest, realizing that most of such circumstances could be treated by separating the experiment in shorter identical blocks with lower numbers of scans. Besides this, we note that peaks that are aliased or folded along an indirect dimension cannot be accurately corrected by
SAFR; a proper designation and a separate treatment of such resonances would be necessary.

In our experience, highly resolved [1]H-detected 2D and 3D spectra benefit from the use of the drift correction the most. Indeed, SAFR significantly increases the available amount of the precious spectrometer time. Multidimensional NMR experiments can be safely recorded even during the helium or nitrogen refill. Although designed specifically for cases known to cause field changes, SAFR can be safely and without additional assumptions used even when no drifts are anticipated to en-
sure unbiased results of the experiment.

## 6 Code availability

All the pulse programs listed in Table S1 in the Supplement and the AU program safrcorr are freely available at https://doi.org/10.5905/ethz-1007-453 (Římal, 2021) under the Apache License, Version 2.0. Details about the composition of the programs and a typical experimental workflow are described in the Supplement.

## 7 Data availability

The experimental data are available from the authors upon request.



## 8      Supplement link

*the link to the supplement will be included by Copernicus*

## 9      Author contribution

VŘ, TW, ME, AB, and BHM contributed to the concept of the method. VŘ performed, analysed, and visualized the experiments and wrote the pulse and AU programs. MC and AAM carried out the experiment on Rpo4/7*. VŘ, MC, AAM, and TW tested the software. RC expressed the ASC and HET-s(218–289) protein fibrils and AT the Rpo4/7* protein complex. TW analysed the field drift of the 1.2 GHz magnet. BHM conceived and supervised the project. VŘ and BHM prepared the original draft. All authors reviewed and edited the article.

## 10      Competing interests

ME is an executive editor and AB and BHM are members of the editorial board of Magnetic Resonance. The peer-review process was guided by an independent editor, and the authors have also no other competing interests to declare. The other authors declare that they have no conflict of interest.

## 11      Financial support

The research has been supported by an ERC Advanced Grant (BHM, grant number 741863, Faster), by the Swiss National Science Foundation (BHM, grant numbers 200020_159707 and 200020_188711), an ETH Research Grant (TW, grant number ETH-43 17−2), the French Agence Nationale de Recherches sur le Sida et les hépatites virales (ANRS, ECTZ71388 & ECTZ100488), and the CNRS (CNRS-Momentum 2018), the LABEX ECOFECT (ANR-11-LABX-0048) within the Université de Lyon program Investissements d'Avenir (ANR-11-IDEX-0007).

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
