# Peer review of "Correction of field instabilities in biomolecular solid-state NMR by simultaneous acquisition of a frequency reference"

_Magnetic Resonance, 2021_

## Author Response (AR1)

**Authors' response to the reviewers' comments**

**Referee #1**

We thank the Referee #1 for careful reading of our manuscript and constructive suggestions. After thorough consideration of all the comments and questions, including the minor ones, we have modified the article as follows:

> ln 70 - The introduction acknowledges other simultaneous acquisition schemes, but a discussion of these existing approaches should be included. What is SAFR contributing that improves on what is already available?

We made it more clear that the cited schemes using multiple acquisitions serve a different purpose than SAFR:

Instead of "Simultaneous acquisition of several free-induction decays (FIDs) in one scan, as in SAFR, is becoming increasingly popular (Gallo et al., 2019; Gopinath and Veglia, 2020; Stanek et al., 2020; Kupče and Claridge, 2017; Sharma et al., 2016)." we write now: "Simultaneous acquisition of several free-induction decays (FIDs) in one scan is becoming increasingly popular **for efficient combination of several multidimensional spectra into one pulse program** (Gallo et al., 2019; Gopinath and Veglia, 2020; Stanek et al., 2020; Kupče and Claridge, 2017; Sharma et al., 2016)**, while in SAFR, the additional FID serves for the frequency calibration of the main experiment**."

> ln 75 - I thought Eq 1 was clear in how it uses t_acq, but the sentence that follows introduces relationships with t_dir, t1, and t2 in various cases. I assume t_acq is the time coordinate along the FID acquisition, and maybe t_dir is the total time it takes to collect a FID? You need explicit definitions in this section.

We are particularly grateful for pointing this out, because it deserves a better explanation, indeed. The time coordinate $t_{acq}$ is a general symbol that is later replaced by $t_{dir}$ and $t_{indir}$, which have the same meaning as $t_{acq}$, but are specific for the direct and indirect dimensions, respectively. The paragraph below Eq. (1) now reads:

**"In a multidimensional experiment, Eq. (1) applies both to direct and indirect acquisition with a general time coordinate $t_{acq}$.** For the direct acquisition of an FID, **we will further use the specific symbol $t_{dir}$ for the time $t_{acq}$** (in a 2D typically $t_2$). Any indirect evolution block **will be emphasised by referring to $t_{acq}$ by $t_{indir}$** (in a 2D typically $t_1$)."

and a small change appears in the last sentence above Eq. (2) as well, which has become:

"The phase difference of the time-domain signal of an isotope with a gyromagnetic ratio γ depends linearly on **the time variable $t_{dir}$:**"

> ln 125 - There seems to be a gap between the theory and the application. The theory derives a rotation matrix (and its inverse) that accommodates different rotations for the real and imaginary components, but the paper does not illustrate or completely explain how the SAFR spectra are processed so that the corrections may be applied.

We have added a short subsection **2.4 Application of SAFR**:

"The field changes $\Delta B_k$ for every FID $k$ can be measured by SAFR accompanying a multidimensional experiment. We prepared an AU program "safrcorr", which can be run directly in Bruker Topspin, that applies the theoretical considerations described above to the experimental data. By this program, the raw data (before Fourier transform) are read. The direct dimension is corrected for every FID according to Eq. (3). For each indirect dimension, the phase differences $\Delta \varphi_k'$ are calculated by Eq. (5) and used to correct the data points using Eq. (7). In this way, all the points in the time domain are compensated for the field drift along all dimensions. A new dataset is created, which can be further processed by standard means."

> ln 255 - The drift is clearly corrected, but how robust is SAFR in recovering peak shapes / peak volumes? The uncorrected peaks in the 4 spectra in Figure 5B show incorrect positions, but each has a correct peak shape. The variation in position makes the "summed" representation in the last row appear to have broadening along 1H. Discussing this or quantifying this would be a nice addition.

Since the peak shapes in Fig. 5 are only weakly affected by the drift, we point out this now in the corresponding paragraph. We also discuss the summed spectra more explicitly. Thus, the subsection 4.2 has been extended by:

"**The drift mostly affects the positions of the resonances, but the line shapes are distorted as well, visible mainly in experiments 2 and 3 in Fig. 5 (b). Both the positions and the shapes of the peaks are restored after the correction, such that possible differences to a control experiment remain negligible (Fig. S12 in the Supplement). Moreover, the summed spectra in Fig. 5 (a) and (b) emphasize the peak broadening that would be caused by the spectral summation without the drift correction and, at the same time, that SAFR doesn't require chemical-shift calibration of individual spectra when the whole progress of the field drift as in Fig. 5 (c) is known.**"

> ln 266 - The delay time between scans along the indirect dimension of the outermost loop will be much longer than between scans along inner loop dimensions. While SAFR appears to work great, is there some additional benefit that could result from either optimizing the order of the indirect dimensions or running multiple experiments and varying the dimension order in each (in all cases, using the same total number of transients as the original)?

The suggested treatments could in principle improve the uncorrected spectra to some extent. On the other hand, we think that there would be no additional gain in the peak shapes after the correction, because SAFR should be independent on the loop order and spitting of the experiment into shorter runs. We have included a discussion of these approaches and we have added a statement about the robustness of SAFR in subsection 4.3:

**"The strongest influence of the drift is seen along the $^{13}C$ axis, which corresponds to the evolution time that is incremented stepwise after a particular NH plane (122 rows) is acquired, i.e., $^{13}C$ belongs to the outermost loop in the pulse program. Without SAFR, one could consider that, in ppm units, the resonances are narrower along the $^{13}C$ dimension than $^{15}N$ (1.1 ppm and 1.0 ppm line widths in $^{13}C$ compared to 1.4 ppm and 2.0 ppm in $^{15}N$ for the two peaks shown in Fig. 6, respectively), which makes $^{13}C$ more sensitive to the effects of the field drift. Therefore, the uncorrected spectrum would in principle slightly profit from exchanging the order of the loops. Whereas this and other alternative approaches, such as splitting the experiment into several blocks with smaller number of transients and varying the dimension order, would bring only minor improvements to the uncorrected spectrum but would still lead to line broadening after averaging similarly as in Fig. 5 discussed above, SAFR and the subsequent spectral correction are independent on these technical adjustments of the pulse program and yield the same final results."**

> ln 63 - "Our work presented here extends the linear drift compensation (Najbauer and Andreas, 2019) for a general non-linear case." You may be underselling yourself. In addition to handling nonlinear drift, you are also handling discontinuities that arise from things like the helium fill shown in Fig 5C.

Although we are pleased by this comment and would be happy to include a stronger statement in this part, we would rather stay careful about correcting discontinuities, which present one of the limitations of the SAFR method. Significant field changes that occur during the acquisition of one FID cannot be fully compensated for by SAFR because (i) we assume that the field is constant during one FID as expressed above Eq. (2) and (ii) the reference spectrum might be acquired at a different field than during the main FID. We only slightly expanded the quoted sentence:

"Our work presented here extends the linear drift compensation (Najbauer and Andreas, 2019) for a general non-linear case **with no strict assumptions on the expected progress of the field over time including e.g. the changes appearing during a helium fill**."

> ln 130 - Fig 1 is a very nice visual aid. I found it slightly distracting that the arc segments depicting both rotations and all dashed lines "overshoot" what they are labeling.

We have modified Fig. 1 accordingly and we agree that it looks better now:

[Figure]

> ln 345 - The statement of code availability is fine, but why is the data only "upon request?" Can the data be deposited with the code?

There was no particular reason not to publish the experimental data. We have uploaded all the spectra to a freely accessible location (separately from the code) under the Creative Commons Attribution 4.0 License (CC-BY). The section Data availability now reads:

"The experimental data are available **at https://doi.org/10.3929/ethz-b-000522147 (Římal and Meier, 2021).**"

and the references include the following item:

"**Římal, V. and Meier, B. H.: Experimental Data for Correction of Field Instabilities in Biomolecular Solid-State NMR by Simultaneous Acquisition of a Frequency Reference, ETH Zurich [data set], https://doi.org/10.3929/ethz-b-000522147, 2021.**"

We hope that we have sufficiently addressed all the referee's comments.

**Referee #2**

We thank referee #2 for careful reading of our manuscript and for the comment. We assume that there are no changes to the manuscript needed.

**Code update**

Besides the issues raised by the reviewers, we have discovered a bug in the pulse programs that was preventing proper work of the DUMAS experiment on Avance III spectrometers. Therefore, **we updated the pulse programs**. The new link to the code is not available yet and will be supplied in the final version of the manuscript.

---

## Author Response (AR2)

The updated link for the program code is now provided.